# Evaluating Kinetics of Convection Drying and Microstructure Characteristics of Asian Seabass Fish Skin without and with Ultrasound Pretreatment

**DOI:** 10.3390/foods12163024

**Published:** 2023-08-11

**Authors:** Mohammad Fikry, Soottawat Benjakul, Saleh Al-Ghamdi, Mohamed Tagrida, Thummanoon Prodpran

**Affiliations:** 1Department of Agricultural and Biosystems Engineering, Faculty of Agriculture, Benha University, Moshtohor, Toukh 13736, Egypt; moh.eltahlawy@fagr.bu.edu.eg; 2International Center of Excellence in Seafood Science and Innovation, Faculty of Agro-Industry, Prince of Songkla University, Hat Yai, Songkhla 90110, Thailand; soottawat.b@psu.ac.th (S.B.); m.tagridaa@gmail.com (M.T.); 3Department of Food and Nutrition, Kyung Hee University, Seoul 02447, Republic of Korea; 4Department of Agricultural Engineering, King Saud University, P.O. Box 2460, Riyadh 11451, Saudi Arabia; sasaleh@ksu.edu.sa; 5Center of Excellence in Bio-Based Materials and Packaging Innovation, Faculty of Agro-Industry, Prince of Songkla University, Hat Yai, Songkhla 90110, Thailand

**Keywords:** ultrasound, drying, fish skin, mathematical modeling, microstructure

## Abstract

Convection drying in combination with ultrasound pretreatment has emerged as a promising technology for seafood manufacturing. The primary objective of this research was to model the mass transfer process of Asian seabass (*Lates calcarifer*) fish skin without and with ultrasound pretreatment during convection drying at different temperatures (45, 55, and 65 °C). Additionally, the study aimed to examine the impact of ultrasound pretreatment and temperatures on the drying characteristics and specific energy consumption for drying of Asian seabass fish skin. Seven semi-theoretical models, namely Lewis, Page, modified Page, Vega-Lemus, Verma, Henderson and Pabis, and two-term models, were employed to characterize the moisture transfer process. The results of the study indicated a decrease in the moisture content as the drying time increased at different drying temperatures. Higher drying temperatures were associated with an increased drying rate. Among the mathematical models tested, the modified Page model provided a satisfactory description of the thin-layer drying characteristics of fish skin. Fick’s law of diffusion was utilized to determine the effective moisture diffusivities. Comparing the drying of fish skin without (SS) and with ultrasound pretreatment (US-SS), the drying of the latter generally showed higher D_eff_ values. The temperature dependence of the effective diffusivity coefficient was well described by the Arrhenius-type model. An increase in the drying temperature resulted in an increment of the effective moisture diffusivity. In general, the skin pretreated using ultrasound had a reduced drying time, by up to 28%. Additionally, this approach contributed to an approximate 22% reduction in the specific energy consumption, concurrently enhancing the energy efficiency. The microstructure analysis showed that fresh and dried US-SS samples had a more open structure and higher porosity, in comparison to the corresponding SS samples. These findings contribute to the knowledge on the application of ultrasound as the pretreatment of fish skin before drying and provide valuable insights for the development of potential drying techniques in the seafood industry.

## 1. Introduction

Asian seabass (*Lates calcarifer*) is a popular fish species with a high demand in the food industry. During fillet production of Asian seabass, skins are mainly generated [1], leading to a significant amount of waste. Although the skin of Asian seabass contains valuable components such as collagen and bioactive compounds, making it an interesting material for various applications [2], it is a by-product that is typically not efficiently utilized. If not properly managed, the accumulation of fish skin can lead to increased waste disposal costs, environmental pollution, and potential health hazards. Moreover, the underutilization of fish skin represents a loss of opportunity for value-added product development and economic benefits. To address these issues, there is a growing interest in finding innovative and sustainable solutions for the utilization of fish skin in various industries, such as the food, cosmetic, and pharmaceutical sectors, which can not only reduce waste but also add value to the overall fish-processing industry. Nowadays, crispy fish skin snacks in the Asia region have gained significant popularity among consumers, and fish-processing by-products can be better exploited [3].

Crispy fish skin can be produced utilizing cost-effective and straightforward technology that can be customized to meet the nutritional preferences and desires of the community. Additionally, marine resources can be sustainable. Typically, the process of making crispy fish skin involves washing, soaking in a flavoring agent, adding spices, drying, and frying [4].

The drying process is one of the crispy snacks production stages that plays a crucial role in the preservation and quality maintenance of food products. Drying is a widely employed technique to decrease the moisture content and water activity of foodstuff, aiming to minimize the presence of microorganisms and prevent enzymatic and biochemical reactions [5,6]. As drying processes typically demand substantial energy input, it becomes crucial to address the limitations of the available energy sources. With the world’s energy resources being finite, there is a growing need for drying methods that offer high efficiency while keeping the costs low [7].

In the industry, the ultimate objective is to minimize the production time without incurring additional production costs, thus justifying the investments made. Hence, recently, there has been developing interest in exploring innovative or effective techniques to improve the drying process and enhance the properties of dried food materials [8,9,10,11,12]. One potential technique is the ultrasonic-assisted process, which involves the application of high-frequency sound waves to accelerate drying and induce structural modifications. The ultrasonic process has shown potential in enhancing the drying rate and reducing the drying time of various food products. It can also promote moisture removal by facilitating the diffusion of water molecules from the foodstuff matrix [8,13,14]. In addition, this approach can shorten the drying time [15] and reduce the energy consumption [16]. 

In order to achieve the aforementioned goals of ultrasound technology, it is common to utilize ultrasonic pretreatments in combination with traditional drying methods, such as hot-air, microwave, or infrared drying. This involves immersing the samples in a hypertonic aqueous phase or water before drying. Ultrasonication can help remove moisture from the materials without causing a substantial temperature increase [17]. Therefore, various studies have confirmed the advantages and the positive effect of combining ultrasonic pretreatment with drying techniques [16,18,19,20,21].

For a better understanding and an increased drying efficacy, studies on the moisture transfer kinetics in food products have gained significant attention. Various mathematical models have been employed to investigate the phenomenon of moisture transfer, which can be broadly categorized into three types: theoretical, semi-theoretical, and empirical models. Several studies [9,16,22,23,24,25,26,27,28,29,30] have successfully utilized semi-theoretical models to analyze the drying kinetics of different food materials. These models offer a balance between the theoretical accuracy and practical applicability, allowing for efficient analysis of drying procedures. In contrast, theoretical thin-layer equations and assumptions regarding the geometry of typical food often overlook factors such as mass diffusivity and conductivity [31].

Therefore, understanding the mechanisms and dynamics of moisture transfer during drying is essential for optimizing the drying process and preserving the quality of the final product [30]. By carefully managing the drying conditions and employing appropriate drying techniques, it is possible to achieve the desired moisture reduction and ensure the stability and shelf-life of crispy fish skin snacks.

Ultrasonic technology has been implemented for seafood drying, and its impact on drying kinetics, energy consumption, and product quality has been evaluated. Ozyalcin and Kipcak [1] compared ultrasonic pretreatment (US) with microwave drying (MW) and thin-layer infrared (IR) drying of *Loligo vulgaris* (squid). Ultrasonic pretreatment significantly reduced the drying times, and the lowest time was observed when MW drying was applied. Vacuum and ultrasonic pretreatments on *Cancer pagurus* (brown crab) were comparatively examined. Both methods shortened the drying time, thus enhancing the drying effectiveness and lowering the costs [2]. Dehydrated shrimp subjected to ultrasound pretreatment showed a reduced dehydration time by up to 15.6% [3]. Ultrasound-assisted vacuum drying of Mediterranean or black mussels (*Mytilus galloprovincialis*) demonstrated a shortened drying period and an increased effective moisture diffusivity, resulting in low energy consumption [4]. Power ultrasound application during low-temperature drying of salted cod increased the drying rate and showed an average 74% increase in effective diffusivity [5]. Additionally, ultrasound-assisted osmotic pretreatment reduced the microwave freeze-drying time of sea cucumbers by approximately 2 h [6]. 

Frying is a common method to cook seafood, yielding the products with the typical characteristics, especially texture and flavor. The impact of air frying and vacuum frying on the mass transfer and texture of fried fish skin was examined [7]. More recently, several methods, such as continuous frying, air frying, rotary baking, and infrared radiation, etc., have been implemented to produce fried tilapia skin snacks [8]. Wang et al. [8] and Sari et al. [9] evaluated the physicochemical and sensory properties of tilapia skin chips using different techniques. Moreover, the physicochemical characteristics and volatile flavor of fried tilapia skins under three frying methods were investigated [10]. However, to the best of the authors’ knowledge, the effect of ultrasound pretreatment before convection drying at different temperatures on the drying characteristics, specific energy consumption, and microstructure of Asian seabass fish skin has not yet been investigated. Since the research available on describing the drying behavior and comparing the energy consumption of convective drying with and without the assistance of ultrasonic pretreatment for drying seafood is limited, the current study is dedicated to addressing this specific research gap, for Asian seabass fish skin in particular. Hence, the present study aimed to further explore the application of ultrasonic technology in seafood manufacturing by investigating the mass transfer process of Asian seabass (*Lates calcarifer*) fish skin during convection drying at different temperatures (45, 55, and 65 °C), with and without ultrasound pretreatment. The study also evaluated the impact of ultrasound pretreatment and drying temperatures on the drying characteristics, specific energy consumption, and microstructure of Asian seabass fish skin. Consequently, valuable insights for the development of potential drying techniques in the seafood industry can be obtained. 

## 2. Materials and Methods

### 2.1. Sample Preparation and Sonication Treatment (US)

The frozen skin of Asian seabass fish was acquired from a local fish fillets industry in Hat Yai, Thailand. To prepare the skin before further treatments, frozen skins were thawed in running tap water and subsequently descaled. The samples were cleaned and cut into uniform stripes with an average thickness of 3.17 mm, to ensure consistent processing, and then carefully packed in polyethylene bags. The prepared stripes were divided into two groups: control samples and ultrasonicated samples. The fish skin samples were ultrasonicated using an ultrasonic water bath (Elmasonic S 70 (H), Elma Schmidbauer GmbH, Germany), operating at a frequency of 32 kHz, and at input, output, and effective powers of 750, 600, and 150 W, respectively. The ultrasound water bath was equipped with a digital timer and a temperature controller. The laboratory bath had a tank capacity of 6.9 L, with internal dimensions of 50.5 cm (length), 13.7 cm (width), and 10 cm (height). Approximately 4 L of distilled water was added to the bath. The fish skin samples (500 g) were immersed in the water bath, and the temperature was monitored using a thermometer. The bath temperature was maintained at 30 °C, and the treatment duration was set for 30 min. Skins with ultrasound pretreatment and those without pretreatment were named as ‘US-SS’ and ‘SS’ samples, respectively.

Both US-SS and SS samples were stored in a refrigerator at 4 °C prior to being subjected to the drying process. Before proceeding with the experiments, the initial moisture content was determined using a moisture analyzer (MA35, Sartorius, Goettingen, Germany). Both control and ultrasonicated Asian seabass fish skin samples had an average initial moisture content of 1.6 ± 0.12 (g water/g dry matter).

### 2.2. Drying Procedure

A convection oven (Memmert, Schwabach, Germany), with a 1600 W heating unit and a controlled temperature chamber capable of maintaining specific drying temperatures (45 °C, 55 °C, and 65 °C), was used for conducting the drying process. The oven was preheated prior to drying at the proposed temperature for a duration of 10 min for each experiment. For the drying process, roughly 20 ± 2 g of each fish skin sample was used. Throughout the drying process, weight measurements were recorded 3 times at regular intervals of time: every 15 min at the beginning of the process, then every 30 min, and finally, every 60 min, until the final moisture content of 0.07 (g/g d.b.) and a_w_ < 0.6, which were the target limits, were achieved. These limits were chosen as they fall within the acceptable range for achieving dried and microbiologically safe fish products [4,32,33,34]. During drying, the weight loss values of the samples were recorded, and then moisture contents were determined on a dry basis (g water/g dry matter). The drying rate (g water/g dry matter. h) was calculated based on the change in the moisture content over time [27]. For comparing different dried samples, Figure 1 depicts photos of Asian seabass fish skin without and with ultrasound pretreatment before and after drying at various temperatures, via the convection drying method. 

### 2.3. Determination of Moisture Content (MC) and Water Activity (a_w_)

The initial moisture content of the fish skin was measured using an infrared moisture analyzer (MA35, Sartorius, Goettingen, Germany), following the method described by Fikry, et al. [27]. Additionally, the water activity (a_w_) was determined at 25 °C using an Aqua Lab instrument (AquaLab Pre, Decagon, QTEtech, Hanoi, Vietnam). The recorded outcomes represent the average of three replicates [25].

### 2.4. Drying Kinetics Modeling

The determination of the moisture ratio (MR) for fish skin involved the conversion of the acquired experimental drying data [9], as follows:(1)$$\;\mathrm{MR}\;=\frac{M_{t}-M_{e}}{M_{i}-M_{e}}\;$$

The symbols *M_t_*, *M_i_*, and *M_e_* represent the moisture content at time t, the initial moisture content, and the moisture content at equilibrium, respectively.

In theory, the equilibrium state can be achieved after an infinitely long drying period. Consequently, the moisture content at equilibrium (*M_e_*) can be disregarded for extended drying durations, and is comparatively small when compared to both the initial moisture content (*M_i_*) and the moisture content at a specific time (*M_t_*) [28,35]. Therefore, the dimensionless quantity of moisture ratio (MR) can be simplified using Equation (2):(2)$$\mathrm{MR}\;=\frac{M_{t}}{M_{i}}$$
foods-12-03024-t001_Table 1Table 1Thin-layer drying models used for drying of Asian seabass fish skin.ModelMathematical Equation *ReferencesLewisMR = exp (−*k . t*)[36]PageMR = exp (−*k . t^n^*)[37]Modified PageMR = exp (−(*k . t*)*^n^*)[38]Henderson and PabisMR = *a .* exp (−*k . t*)[39]Vega-LemusMR =a + k . t2[40]Verma et al.MR = a.exp− k .t+1− a exp b .t[41]Two-termMR = a .exp−k .t+ b.exp−k1 . t[42]* The empirical constants and coefficients in drying models are denoted by the symbols a, b, k, and n. The variables MR and t represent the moisture ratio and drying time, respectively.


### 2.5. Determination of Drying Characteristics

#### 2.5.1. Effective Moisture Diffusivity

The estimation of effective moisture diffusivity ($D_{eff}$) from drying curves involves the application of Fick’s second law of diffusion. For one-dimensional slab geometry, the equation is given as described by Manzoor, et al. [28]:



(3)
$$\frac{\partial M}{\partial t}=D_{eff}\left(\frac{\partial^{2}M}{\partial r^{2}}+\frac{\eta }{r}\frac{\partial M}{\partial r}\right)$$



In the provided equation, the constant *η* is equal to 0 for planar geometry. The initial and boundary conditions are defined as *M*(*r*,0) = *M* at *t* = 0, and *r* denotes the spatial coordinate, ranging from 0 to *L*, where *L* denotes the half-thickness (m).

This equation assumes a uniform initial moisture distribution and considers moisture migration to occur solely through diffusion. Shrinkage, temperature variations, and external resistance are considered negligible during the drying process [43]. The fish skin samples were treated as having an infinite slab geometry, resulting in the following Fick’s diffusion equation:(4)$$\mathrm{MR}\;=\frac{M_{t}-M_{e}}{M_{e}-M_{i}}=\frac{8}{\pi^{2}}\sum_{n=0\;}^{\infty }\frac{1}{{\left(2n+1\right)}^{2}}\exp\left(-\frac{{\left(2n+1\right)}^{2\;}\pi^{2}}{4L^{2}}D_{eff}t\right)$$

For the first three series, Equation (5) can be expanded as follows:(5)$$\mathrm{MR}=\frac{8}{\pi^{2}}\left(\exp^{-{\left(\frac{\pi }{2}\right)}^{2}N_{Fi}}+\frac{1}{9}\exp^{-9{\left(\frac{\pi }{2}\right)}^{2}N_{Fi}}+\frac{1}{25}\exp^{-25{\left(\frac{\pi }{2}\right)}^{2}N_{Fi}}\right)$$
where *N_Fi_* represents the Fourier number, defined as $\frac{D_{eff}\times t}{4L^{2}}$, with *L* representing half the thickness of the material. In the case of long drying times, when the Fourier number exceeds 0.1 and the unaccomplished moisture ratio is less than 0.6, the first term of the series becomes dominant. Consequently, the equation can be simplified to Equation (6) [44]:(6)$$\mathrm{MR}\;=\frac{8}{\pi^{2}}\exp\left(-\frac{\pi^{2}\;D_{eff}t}{4L^{2}}\right)$$
where *D_eff_* represents the effective moisture diffusivity (m^2^/s), *L* denotes the sample thickness (m), and *t* represents the drying time (s).

The aforementioned equation can be evaluated in terms of the Fourier number, denoted as *F_o_*, which has a numerical value of $\frac{D_{eff}\times t}{4L^{2}}$ [44]. Equation (7) can be expressed as follows: (7)$$\mathrm{MR}\;=\frac{8}{\pi^{2}}\exp\left(-\pi^{2\;}F_{o}\right)$$

Therefore,
(8)$$F_{o}=-0.101\ln\left(\mathrm{MR}\right)-0.0213$$

Effective moisture diffusivity can be determined using the following equation:(9)$$D_{eff}=\frac{F_{o}}{\frac{t}{4L^{2}}}$$

Alternatively, the slope method can be adopted to determine the effective moisture diffusivity ($D_{eff}$). This method involves plotting the natural logarithm of the moisture ratio (ln(MR)) against the drying time (t). The resulting plot forms a straight line, and the slope of this line corresponds to the value of $D_{eff}$ [28]:(10)$$Slope=\frac{\pi^{2}D_{eff}}{4L^{2}}.\;$$

#### 2.5.2. Calculation of Activation Energy

The activation energy (*E_a_*) was calculated using the Arrhenius equation, as depicted in Equation (11), which establishes a relationship between the effective moisture diffusivity ($D_{eff}$) and temperature. *E_a_* can be obtained from the slope of ln(*D_eff_*) vs. 1/*T* plot [28].
(11)$$D_{eff}=D_{o}\exp\left(-\frac{E_{a}}{RT}\right)$$
where $D_{o}$ is defined as the Arrhenius or pre-exponential factor (m^2^/s), *E_a_* represents the activation energy (J/mol), *R* is the universal gas constant (8.314 J/mol·K), and *T* indicates the absolute drying temperature (K). The equation is rearranged as follows:(12)$$\ln\left(D_{eff}\right)=\ln\left(D_{o}\right)-\left(\frac{E_{a}}{RT}\right)$$

### 2.6. Determination of Energy Consumption during Drying

#### 2.6.1. Specific Energy Consumption (SEC)

The energy consumption by the convection dryer was determined using Equation (13) [45]. Equation (14) defines the energy consumption for the ultrasonic device [9,46].
(13)$$E_{CD}=P_{CD}.t$$
(14)$$E_{US}=U.\;I.\;Cos\;\emptyset \;.\;t$$

In the aforementioned equation, the variables $E_{CD}\;$ and $E_{US}$ denote the energy consumed by the oven and the ultrasonic device (kW h), respectively, $P_{CD}\;$ represents the output power of the oven (1.6 kW), U, *I*, and Cos ∅ are the applied voltage (V), current (A), and power factor (0.8) of the ultrasound device, respectively, and *t* represents the treatment time (hours).

To calculate the total energy consumption for ultrasound pretreated skin (US-SS), the energy consumption of the ultrasonic device ($E_{US}$) was added to the energy consumption of the convection dryer ($E_{CD})$ [45].

The specific energy consumption (SEC, MJ/kg_water_) required for drying the fish skin was calculated according to Equation (15) [47]: (15)$$SEC=\frac{\;E_{T}}{X_{s}\;\left(M_{i}-M_{f}\right)}$$
where *E_T_* is the total energy consumed by the device (kW h), $X_{s}$ refers to the dry mass of fish skin (kg), while $M_{i}$ and $M_{f}$ are the initial moisture content and the final moisture content (kg/kg d.b.), respectively.

#### 2.6.2. Energy Efficiency

The energy required for vaporizing water from a drying substance surpasses that needed for surface water evaporation [48]. At temperatures of 45, 55, and 65 °C, the latent heat of water vaporization was computed as 2390, 2368.3, and 2345.67 kJ/kg, correspondingly. The energy efficiency was determined by Equation (16) [47,49]:(16)$${ή}_{e}=\frac{E_{eva}}{\;E_{T}}=\frac{h_{fg\;}\;m_{w}}{\;E_{T}}\times 100\;.\;$$
where $\;{ή}_{e}\;$ refers to the energy efficiency (%), *E_eva_* is the energy required to evaporate moisture (kJ), $h_{fg}\;$ represents the latent heat required for water vaporization at the drying temperature (kJ/kg), and $m_{w}$ corresponds to the mass of water that has been removed during the drying procedure (kg). 

In Equation (16), the value of $h_{fg}$ can be determined by utilizing Equation (17) [46,50], where it is expressed as a function of the absolute temperature (*T*, in Kelvin), as demonstrated below:(17)$$h_{fg}=\left(7.33\times 10^{6}-16\;T^{2}\right)\;0.5\;\;;\;\;\;\;\;273.16<T<338.72$$

### 2.7. Microstructure Analysis

The procedure detailed by Başlar, et al. [51] was adopted. The microstructures of fresh and dried fish skin samples were analyzed using a field-emission scanning electron microscope (SEM) (EVO LS 50, Zeiss, Feldbach, Switzerland). Prior to analysis, the samples were cut and fixed using 3% glutaraldehyde. Thereafter, the glutaraldehyde was removed, and the samples were dehydrated using the series of ethanol. Subsequently, the critical drying point (CDP) treatment was applied on the dehydrated skin samples, which were eventually fixed on the SEM stub. A thin layer of gold was applied to the samples to create a reflective surface for the electron beam. Subsequently, the gold-plated samples were visualized using the microscope at an operating voltage of 5 kV.

### 2.8. Data Analysis

The collected data were subjected to two-way analysis of variance (ANOVA). To determine the differences between the means, a least significant difference test was employed with a 95% confidence interval (*p* < 0.05). 

To identify the most suitable drying equation, the drying data for fish skin were fitted to six distinct thin-layer drying equations (Table 1). Non-linear regression analysis of the experimental data was conducted using SPSS software v. 23. The selection of the optimal mathematical models for the drying data was based on several criteria, including the highest coefficient of determination (R^2^), the lowest Chi-square ($X^{2}$), and the lowest root mean square error (RMSE), as shown in Equations (18)–(20) [29,52,53]:(18)$$R^{2}=1-\frac{\sum_{i=1}^{N}{\left(MR_{exp,i}-MR_{pred,i}\right)}^{2}}{\sum_{i=1}^{N}{\left(MR_{exp,i}-\bar{MR_{exp,i}}\right)}^{2}}\;\;\;\;$$
(19)$$X^{2}=\overset{N}{\underset{i=1}{\sum }}\left(MR_{exp,i}-MR_{pred,i}\right)/N-n$$
(20)RMSE=   1N ∑i=1NMRexp,i−MRpred,i2 

The moisture ratio, experimentally observed for a specific instance (*i*), is denoted as $MR_{exp,i}$. The predicted moisture ratio for the same instance (*i*), using the previous instance *N*, is represented as $MR_{pred,i}$. The parameters *N* and *n* correspond to the number of observations and constants, respectively.

## 3. Results and Discussion

### 3.1. Drying Curves and Water Activity of Asian Seabass Skin

Figure 2, Figure 3, Figure 4 and Figure 5 depict the behavior and characteristics of convection drying for Asian seabass skin without (SS) and with ultrasound pretreatment (US-SS). Figure 2 illustrates that as the drying time increased, the moisture content of Asian seabass fish skin decreased for each temperature (45 °C, 55 °C, and 65 °C), indicating the effectiveness of the drying process. Furthermore, Figure 2 demonstrates that the desired moisture content of 0.07 (g water/g dry matter) for SS dried under convection was obtained within approximately 10 h, 6.85 h, and 3.75 h, at drying temperatures of 45 °C, 55 °C, and 65 °C, respectively. In comparison, the target moisture content of 0.07 (g water/g dry matter) for dried US-SS was achieved at around 8.42 h, 5.67 h, and 2.92 h when the same drying temperatures were used. These results provide clear evidence that the ultrasound pretreatment before convection drying reduced the drying time by approximately 19–28%. This finding aligns with the results reported by Gong, et al. [11] and Liu, et al. [54], where the application of a 30-min ultrasonic pretreatment resulted in shorter drying times, compared to untreated samples.

Similarly, Figure 3 demonstrates that as the drying time (t) increased, the moisture removal percentage decreased for both high and low drying temperatures. It was evident that higher drying temperatures generally resulted in higher moisture removal at the same drying time. In general, higher temperatures facilitate faster moisture removal, leading to a greater overall moisture reduction within a given drying duration. Similar outputs were found by Kocabay [9].

Figure 4 reveals that the drying rate of seabass fish skin decreased as the drying time increased for both SS and US-SS, regardless of the drying temperature. Similarly, Figure 5 demonstrates that the drying rate gradually decreased from the initial moisture content to a lower value as the drying progressed. These trends were in agreement with the results obtained by Darvishi, et al. [55] and Mohd Rozainee and Ng [56].

Table 2 presents the water activities of Asian seabass skin subjected to drying in a convection dryer without (SS) and with ultrasound pretreatment (US-SS). The findings indicated that the water activity of the dried fish skin ranged from 0.530 to 0.586, falling within the safe limits for microbiological concerns (a_w_ < 0.6), which was confirmed by Chandra, et al. [34]. As depicted in Table 2, the water activity values of US-SS samples at various temperatures were slightly lower than those of SS samples. The decrease in water activity observed in the samples subjected to ultrasonic pretreatment in a convection dryer (US-SS) can be attributed to enhanced mass transfer and reduced microstructural collapse, specifically the preservation of pores and moisture transport microchannels during the drying process. This resulted in a faster rate of moisture migration during the drying process. Similar findings were reported by Chandra, et al. [34] and Fernandes, et al. [57], who also highlighted the effective role of ultrasonic pretreatment in forming microchannels that reduce water diffusion resistance and enhance water diffusion during drying.

### 3.2. Drying Kinetic Modeling

The moisture ratios (MRs) obtained from convection drying of both fish skins, SS and US-SS, were fitted into seven thin-layer drying models, as presented in Table 1. The statistical regression of the models, including the drying model constants and the statistical indicators (R^2^, 𝑋^2^, and RMSE) for judging the goodness of fit, is presented in Table 3. Across all temperatures, the proposed models yielded average R^2^ values ranging from 0.969 to 0.999. The regression analysis indicated that the modified Page model demonstrated the best fit for both SS and US-SS, exhibiting the highest average coefficient of determination, the lowest 𝑋^2^, and the lowest RMSE. Figure 6 illustrates a comparison between the experimental and predicted moisture ratios, which were calculated using the modified Page model. These findings aligned with those obtained by Fikry, et al. [27] and Manzoor, et al. [28]. This implied that the selected model was suitable for accurately describing the drying behavior, optimizing drying processes, estimating drying times, and predicting moisture contents of fish skin under the given experimental conditions.

### 3.3. Temperature Dependency and Activation Energy

Effective moisture diffusivity (D_eff_) is a crucial parameter related to the physical and thermal characteristics of materials, particularly during the drying of food. It represents the rate at which moisture is transported over distances during the drying process. Figure 7 demonstrates that the D_eff_ values exhibited an increasing trend with the rising temperature. Specifically, for drying of SS, the D_eff_ values ranged from 8.66 × 10^−11^ to 2.31 × 10^−10^ m^2^/s, while for drying of US-SS, the D_eff_ values varied from 8.16 × 10^−11^ to 3.22 × 10^−10^ m^2^/s. The effective moisture diffusivity values were within the general range of 10^−12^ to 10^−7^ m^2^/s for food products [29,51]. Therefore, these findings were in line with previously published studies conducted on different food materials [8,9,13,28], validating the observed behavior of moisture diffusion during drying processes. These values indicated the effectiveness of the drying methods at different temperatures. Generally, higher values of D_eff_ indicate faster moisture diffusion, suggesting an improved drying efficiency. Overall, for US-SS, convection drying generally showed higher D_eff_ values, implying enhanced moisture removal capabilities compared to drying of SS.

Activation energy (E_a_) is the energy required to initiate moisture diffusion in food or the minimum energy needed to start the drying process [53]. The results of this study demonstrated that the relationship between the effective diffusivity coefficient and the drying temperature could be accurately described using the Arrhenius correlation, as depicted in Figure 8. The activation energy and diffusion consistency coefficient (D_o_) were determined by employing the relationship between ln (D_eff_) and the inverse of the absolute temperature, as described by Equations (21) and (22). The statistical analysis, based on the Arrhenius-type equation, yielded activation energy values of 43.67 kJ/mol for SS and 61.26 kJ/mol for US-SS samples. Additionally, the diffusion consistency coefficient values were found to be 1.19 × 10^−3^ m^2^/s for SS and 9.23 × 10^−1^ m^2^/s for US-SS samples. From previous studies, the activation energy values of drying for different food materials have been reported to fall within the range of 12.7 to 110 kJ/mol [58], and the obtained activation energy values in this study aligned well with those previous findings [9,29].
(21)$$\;D_{\mathrm{eff}}|\mathrm{SS}=1.19\times 10^{-3}exp\left(-\frac{43675}{RT\;}\right)\;\;\;\;\;\;\left(R^{2}=0.923\right)$$
(22)$$D_{\mathrm{eff}}|\mathrm{US}-\mathrm{SS}=9.23\times 10^{-1}exp\left(-\frac{61256}{RT\;}\right)\;\;\;\;\;\;\left(R^{2}=0.997\right)$$

### 3.4. Specific Energy Consumption (SEC)

Reducing the energy requirements in the drying process of fish skin is a crucial consideration, as it leads to lower environmental problems and operation costs [59]. To evaluate the production cost of dried fish skin, the energy consumption for drying of SS and US-SS at different drying temperatures was calculated.

A comprehensive depiction of the total energy consumption and energy efficiency is presented in Figure 9. According to Figure 9a, the specific energy consumption values for convection drying of SS varied from 38 to 98 MJ/kg_water_. For convection drying of US-SS, the values ranged between 30 and 85 MJ/kg_water_. Notably, the drying time significantly decreased as the temperature increased, resulting in a reduced specific energy consumption (Table 3). As anticipated, an increase in temperature led to a higher thermal gradient and improved mass transfer, resulting in quicker moisture evaporation. Consequently, the drying time was reduced, leading to a decrease in SEC [60]. The higher drying temperature led to lower energy consumption, likely due to a shorter processing time at higher temperatures. Among the different drying temperatures, the lowest energy consumption was observed at 65 °C. Therefore, selecting a higher drying temperature is crucial for reducing the energy cost. 

The impact of the drying temperature on the energy efficiency in convective air drying with and without ultrasound pretreatment is portrayed in Figure 9b. The data analysis revealed that as the temperature increased, energy efficiency was enhanced in both scenarios. For instance, when the temperature ranged from 45 to 65°C, the thermal efficiency for SS exhibited an increase from 2.4% to 6.1%. Similarly, the thermal efficiency for US-SS experienced an elevation from 2.8% to 7.9% as the temperature was raised. On the other hand, with the ultrasonication pretreatment within the same temperature range, the thermal efficiency escalated by an average percentage of 22%. These findings were consistent with previous studies [45,51]. Consequently, the convection drying method in combination with ultrasound pretreatment could be effectively applied to dry fish skin, as it reduced the specific energy consumption by an average percentage of 22% compared to the convection drying of skin without any pretreatment. These findings were in agreement with those reported by Abbaspour-Gilandeh, et al. [46].

These findings suggested that a lower energy consumption could reduce the production costs for dried fish skin. By selecting higher drying temperatures, which have a lower energy consumption, producers can decrease their operational expenses, making the process more economically feasible. In addition, adopting the convection drying method in combination with ultrasound pretreatment could be an effective way to further reduce the specific energy consumption in the fish skin drying process. As the specific energy consumption is decreased, there will be a reduction in greenhouse gas emissions and overall energy consumption, contributing to a more sustainable and eco-friendly process.

### 3.5. Microstructure of Dried Fish Skin

The impact of different treatments on the microstructure of SS and US-SS samples was investigated using SEM (Figure 10 and Figure 11). The US-SS samples exhibited a more open structure and higher porosity, with visible wrinkles on the skin surface, compared to the SS samples. Cross-sections of untreated SS samples revealed the presence of some globules, more likely lipid droplets, along with collagen matter. However, no globules were entirely present in the US-treated SS samples. The results indicated the impact of ultrasonication on microstructural changes in SS, as shown by the loss of fat and collagen molecules, likely due to the cavitation behavior of US [61].

In contrast, the dried samples, whether treated or untreated with ultrasonication, showed a decrease in thickness, with the presence of cracks that might have formed during the release of fat globules and other materials. However, no significant difference was observed in these cracks between the different samples. The use of the ultrasonic technique led to less shrinkage of the samples, possibly due to the shorter drying time, faster evaporation of water, and less destruction of some molecules in the skin. Both treatments, US and drying, enhanced the shrinkage with the increasing temperature. This was attributed to the porous structures caused by the loss of oil and other molecules in the fish skin during US.

After drying, the porosity and open structure of the US-SS samples were higher than those of the SS samples. Preservation of porosity and the open structure occurred at low temperatures of drying. Similar findings were reported in other studies [8,62]. Furthermore, the cavitation effect of the US treatment appeared to augment the quantity of materials released from the fish skin. Ultrasound treatment is believed to induce the formation of microchannels and disrupt tissue structures, leading to increased porosity and intercellular spaces, thereby releasing more materials [63]. These changes are thought to facilitate water escape, along with the release of more materials from the skin, such as fats and collagen, thereby reducing the drying time [54].

In the present study, frozen skin was used for drying. Freezing and thawing can determine the drying behavior and drying efficacy in both negative and positive ways. The aggregation of proteins and lipid oxidation might bring about the dense structure of the skin during freezing/thawing, thus lowering the water evaporation. On the other hand, tissue damage caused by ice crystals during freezing might favor the looser structure, thereby facilitating the evaporation of water from the skin.

These outcomes suggested that ultrasonication enhanced water escape during drying, resulting in faster evaporation and a reduced drying time. Consequently, the drying process became more efficient and could lead to higher production throughput.

Additionally, the implications of reduced shrinkage in ultrasonically treated fish skin have significant benefits in terms of product uniformity, handling, packaging, shelf-life, and overall economic viability during storage and transportation. These advantages contribute to a more marketable and sustainable product for the fish skin-processing industry.

## 4. Conclusions

When the convection drying time increased, the moisture content, moisture ratio, and drying rate decreased for both the skin without (SS) and with ultrasound pretreatment (US-SS), regardless of the temperature used. The outcomes revealed that the drying time significantly decreased with the increasing temperature for both SS and US-SS. The dried fish skins, whether SS or US-SS, were determined to be microbiologically safe, with an a_w_ below 0.6. Additionally, at each temperature level, the drying times for US-SS were consistently shorter than those for SS. The drying behavior of fish skin, whether SS or US-SS, was satisfactorily described by the modified Page model. Drying of US-SS exhibited higher D_eff_ values, indicating the higher moisture removal capabilities compared to drying of SS. The dependency of the effective diffusivity coefficient on the drying temperature was accurately predicted using the Arrhenius-type correlation. Drying of US-SS reduced the specific energy consumption and increased the energy efficiency by an average percentage of 22% compared to drying of SS. Therefore, ultrasound pretreatment accelerated the drying process, reduced the energy consumption, and achieved the desired final moisture content in a shorter period for fish skin drying. Dried US-SS samples had a more open structure and higher porosity, compared to their dried SS counterpart.

In summary, the main findings revealed an improved drying efficiency, enhanced sustainability, scientific advancements in drying kinetics, optimization of drying parameters, the potential for innovative seafood products, and the application of ultrasound pretreatment in the seafood industry. These findings contribute to the improvement of drying technologies, providing valuable insights for researchers, industry practitioners, and policymakers aiming to enhance the efficiency and sustainability of food-processing operations.

## Figures and Tables

**Figure 1 foods-12-03024-f001:**
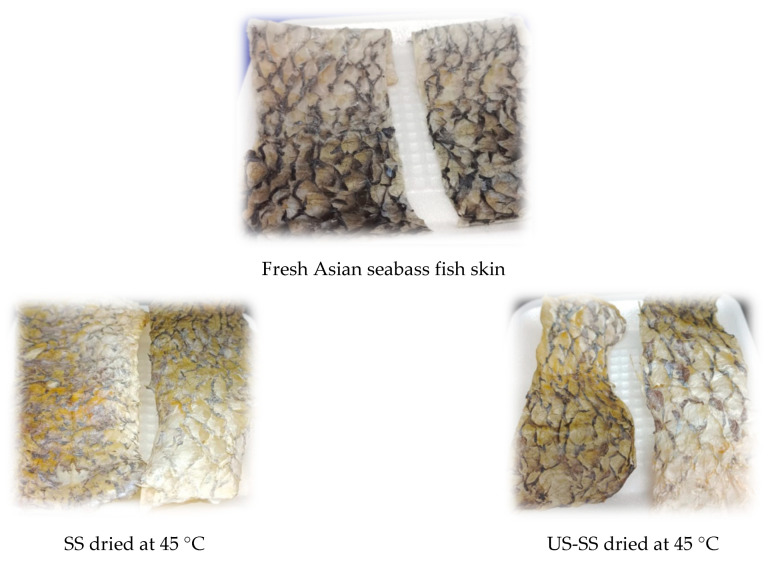
Photos of fresh Asian seabass fish skin and dried skin without (SS) and with ultrasound pretreatment (US-SS) after drying at various temperatures.

**Figure 2 foods-12-03024-f002:**
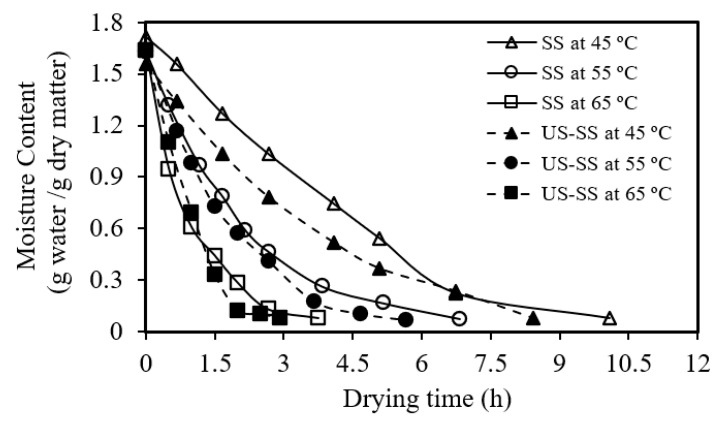
Relationship between the moisture content of Asian seabass fish skin without (SS) and with ultrasound pretreatment (US-SS) and the drying time.

**Figure 3 foods-12-03024-f003:**
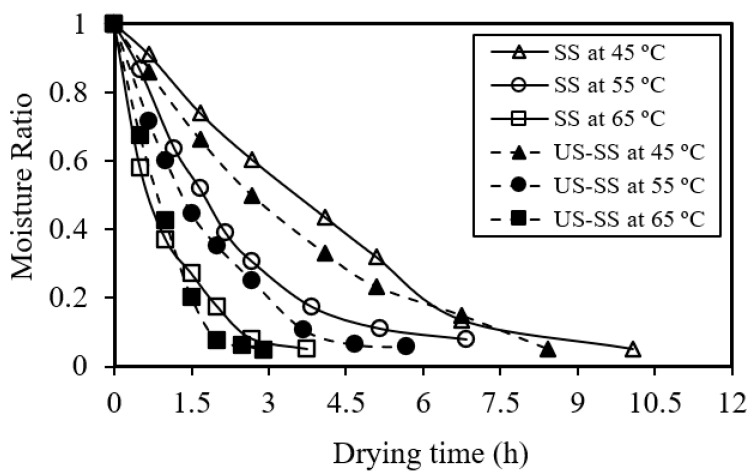
Moisture ratio as a function of the drying time of Asian seabass fish skin without (SS) and with ultrasound pretreatment (US-SS).

**Figure 4 foods-12-03024-f004:**
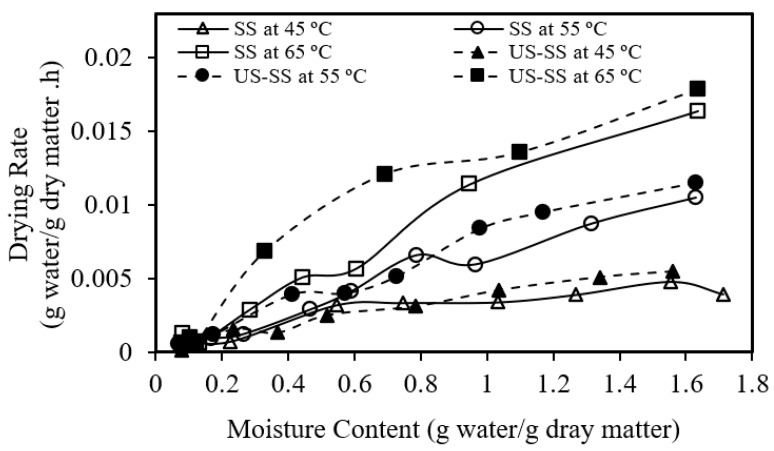
Drying rate versus moisture content of Asian seabass fish skin without (SS) and with ultrasound pretreatment (US-SS).

**Figure 5 foods-12-03024-f005:**
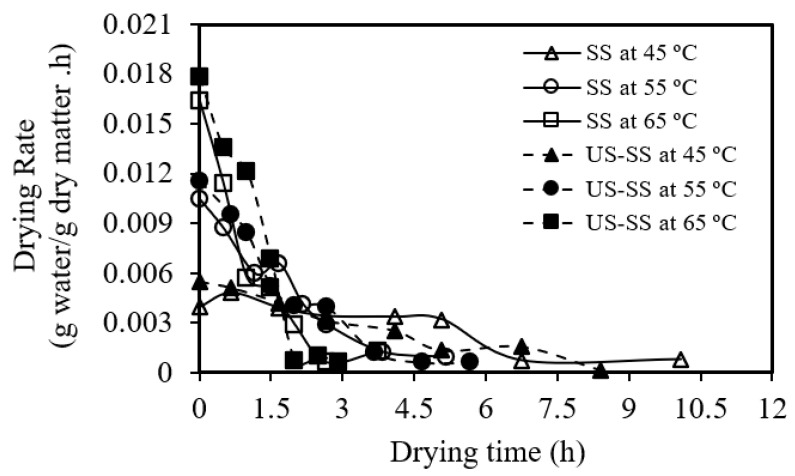
Relationship between the drying rate of Asian seabass fish skin without (SS) and with ultrasound pretreatment (US-SS) and the drying time.

**Figure 6 foods-12-03024-f006:**
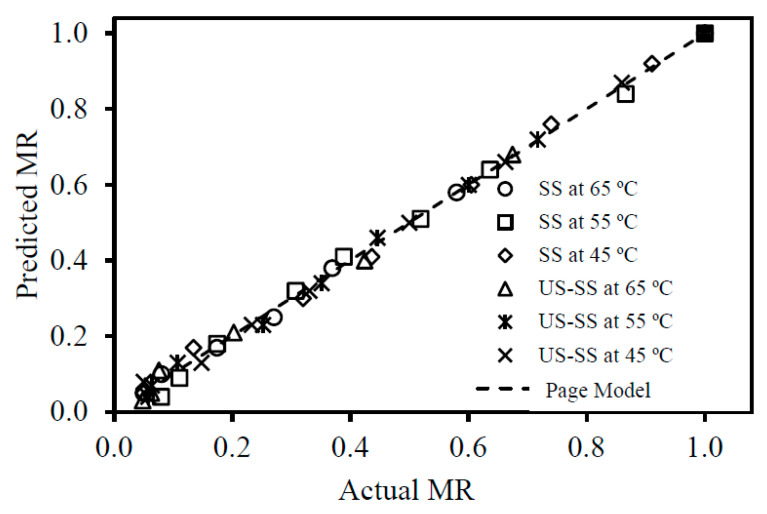
Comparison of experimental vs. predicted moisture ratios of the modified Page model for convection drying of Asian seabass skin without (SS) and with ultrasound pretreatment (US-SS).

**Figure 7 foods-12-03024-f007:**
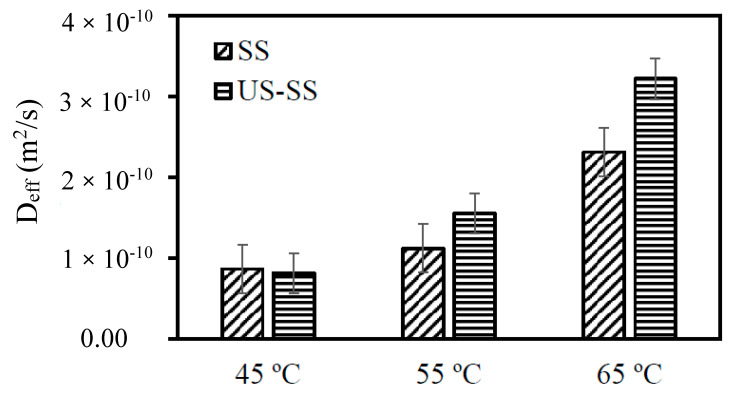
Influence of the drying temperature and ultrasound pretreatment on the effective moisture diffusivity of dried Asian seabass fish skins. SS: without ultrasound pretreatment, US-SS: with ultrasound pretreatment. Error bars represent the confidence intervals at 0.05.

**Figure 8 foods-12-03024-f008:**
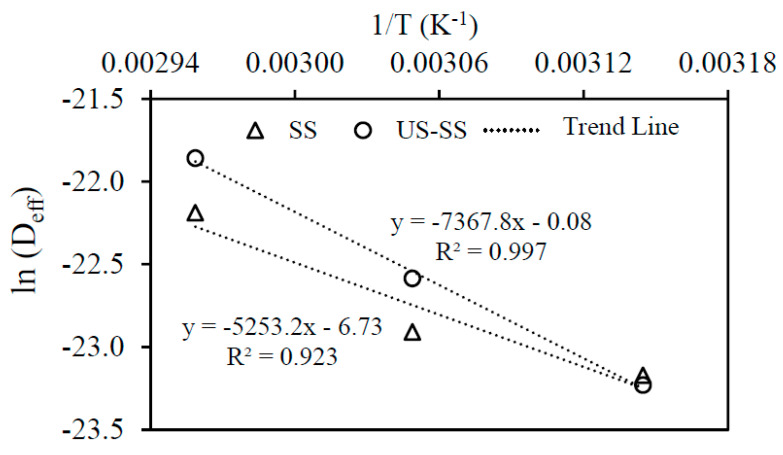
Arrhenius-type relationship for dried Asian seabass fish skin.

**Figure 9 foods-12-03024-f009:**
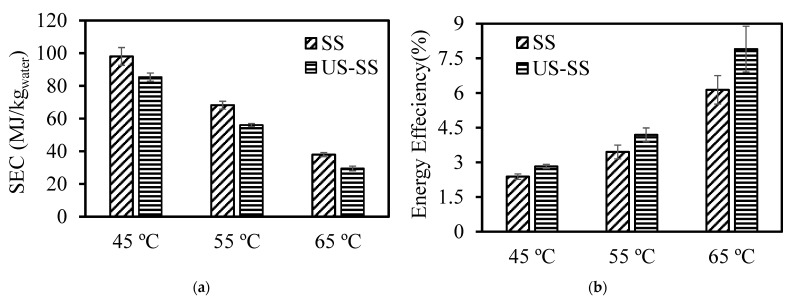
(**a**) SEC (MJ/kg_water_) and (**b**) energy efficiency (%) for Asian seabass skin without (SS) and with ultrasound pretreatment (US-SS), dried at different temperatures. Error bars represent the confidence intervals at 0.05.

**Figure 10 foods-12-03024-f010:**
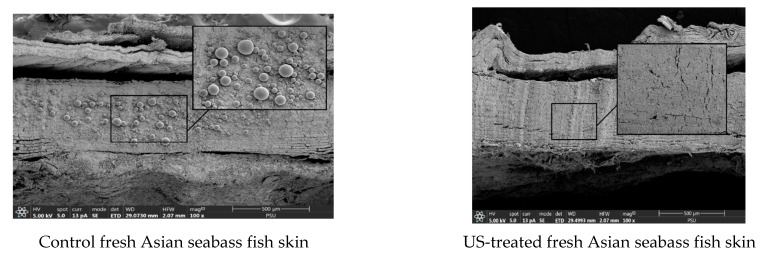
SEM images of cross-sections of SS and US-SS samples with different drying temperatures. Magnification: 200×, with a scale of 500 µm.

**Figure 11 foods-12-03024-f011:**
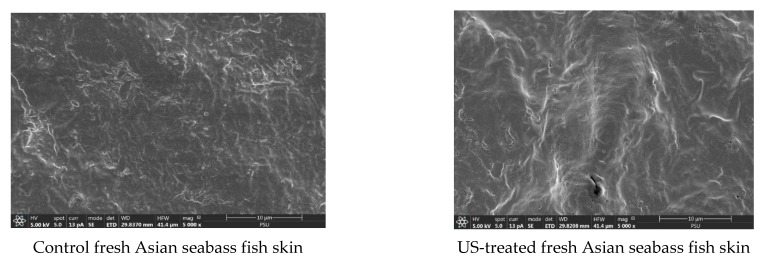
SEM images of the surfaces of SS and US-SS samples with different drying temperatures. Magnification: 5000×, with a scale of 10 µm.

**Table 2 foods-12-03024-t002:** Moisture content (MC), water activity (a_w_), drying time, and specific energy consumption (SEC) of dried Asian seabass fish skins under different treatments.

Responses *	Temperature (°C)	SS (Mean ± Standard Deviation)	US-SS (Mean ± Standard Deviation)
Moisture Content (g/g, d.b.)	45	0.082 ^Aa^ ± 0.02	0.062 ^Aa^ ± 0.01
	55	0.071 ^Aa^ ± 0.01	0.068 ^Aa^ ± 0.02
	65	0.075 ^Aa^ ± 0.01	0.078 ^Aa^ ± 0.01
Water activity (a_w_)	45	0.576 ^Aa^ ± 0.002	0.530 ^Bb^ ± 0.004
	55	0.585 ^Aa^ ± 0.014	0.539 ^Bab^ ± 0.014
	65	0.586 ^Aa^ ± 0.01	0.559 ^Ba^ ± 0.007
Drying time (h)	45	10.08 ^Aa^ ± 0.195	8.42 ^Ba^ ± 0.07
	55	6.83 ^Ab^ ± 0.215	5.60 ^Bb^ ± 0.07
	65	3.75 ^Ac^ ± 0.065	2.92 ^Bc^ ± 0.11
SEC (MJ/kg water)	45	98.02 ^Aa^ ± 4.82	85.39 ^Ba^ ± 2.2
	55	68.2 ^Ab^ ± 2.15	56.05 ^Bb^ ± 0.80
	65	37.97 ^Ac^ ± 1.11	29.50 ^Bc^ ± 1.24

* For each response, the values in the same row or column having the same capital or lowercase letter, respectively, are not significantly different via pairwise LSD tests at a confidence level of 95%.

**Table 3 foods-12-03024-t003:** Coefficients of the proposed models and statistical parameters for thin-layer drying of Asian seabass fish skin without (SS) and with ultrasound pretreatment (US-SS).

Model	Parameters *	SS			US-SS		
		45	55	65	45	55	65
Lewis	k	0.222	0.414	0.951	0.274	0.531	0.98
	R^2^	0.992	0.992	0.994	0.994	0.997	0.995
	X^2^	0.001	0.001	0.001	0.001	0.000	0.002
	RMSE	0.029	0.029	0.024	0.027	0.017	0.041
Page	k	0.139	0.374	0.979	0.228	0.508	0.928
	n	1.327	1.129	0.865	1.139	1.07	1.274
	R^2^	0.995	0.996	0.999	0.999	0.998	0.999
	X^2^	0.0006	0.0007	0.0002	0.000	0.000	0.000
	RMSE	0.0203	0.0219	0.0116	0.014	0.015	0.018
Modified Page	k	0.226	0.419	0.976	0.532	0.938	0.532
	n	1.326	1.115	0.865	1.064	1.290	1.064
	R^2^	0.998	0.998	0.999	0.999	0.999	0.999
	X^2^	0.001	0.001	0.000	0.000	0.000	0.000
	RMSE	0.022	0.022	0.012	0.014	0.019	0.014
Henderson and Pabis	a	1.048	1.032	0.981	1.024	1.013	1.029
	k	0.235	0.431	0.932	0.282	0.539	1.005
	R^2^	0.981	0.994	0.995	0.995	0.998	0.994
	X^2^	0.003	0.001	0.001	0.001	0.000	0.002
	RMSE	0.046	0.028	0.025	0.024	0.017	0.040
Verma	a	−20.933	−24.194	0.849	−29.12	−40.29	−71.30
	b	−0.093	−0.316	−5.112	−0.15	−0.39	−0.55
	k	0.088	0.312	0.807	0.15	0.39	0.55
	R^2^	0.992	0.993	0.999	0.999	0.998	0.997
	X^2^	0.001	0.001	0.000	0.000	0.000	0.0014
	RMSE	0.029	0.027	0.012	0.009	0.016	0.0278
Two-term Exponential	a	0.849	0.000	0.005	0.58	0.506	0.013
	b	0.151	1.035	1.043	0.44	0.507	1.016
	k	0.807	−3.611	0.235	0.28	0.539	1.005
	k_1_	5.113	0.434	0.235	0.28	0.539	1.005
	R^2^	0.999	0.995	0.996	0.995	0.998	0.994
	X^2^	0.0005	0.0019	0.0076	0.002	0.001	0.005
	RMSE	0.0122	0.0232	0.0465	0.024	0.017	0.040
Vega-Lemus	a	1.007	1.001	0.966	0.989	0.985	0.999
	k	−0.088	−0.164	0.312	−0.098	−0.194	−0.357
	R^2^	0.996	0.983	0.969	0.994	0.988	0.998
	X^2^	0.0005	0.0032	0.0044	0.001	0.002	0.001
	RMSE	0.0194	0.0481	0.0561	0.023	0.039	0.028

* a and b refer to the models’ constants (dimensionless), n is the exponent (dimensionless), while, k and k_1_ denote the drying rate constants (h^−1^).

## Data Availability

The data supporting the findings of this study are available from the corresponding author upon request.

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
