# Peer review of "Evaluating Kinetics of Convection Drying and Microstructure Characteristics of Asian Seabass Fish Skin without and with Ultrasound Pretreatment"

_foods, 2023, doi:10.3390/foods12163024_

Round 1
Reviewer 1 Report
Manuscript recieved for revision investigates the combination of ultrasound and convective drying of Asian Seabass Fish Skin, with the modeling mass transfer kinetics and the effects on microstructure characteristics of the samples.
All detailed corrections are noted in manuscripts’ pdf file.
Title of the work is correct and appropriate.
Abstract section needs some corrections.
Introduction section is elaborative and appropriate, but some suggestions are noted.
Material section describes all samples’ preparation, conducted testing and calculations in sufficient details.
Results and discussion is presented in sufficient detail, with proper referencing to other authors’ work. Some specific comment are noted in manuscript pdf file.
Conclusion section is appropriate, and presented conslusions are adequalty derived from presented results.

Author Response
Dear respected reviewer,
Many thanks for your valuable comments and suggestions. The corrections as suggested by the reviewer have been done and for easy track, they are highlighted with yellow color in the manuscript.
- Abbreviation in abstract has been clarified in line (26).
- The phenomenon of moisture transfer in the introduction has been deleted to make the text concise.
- The reference style of Foods was double checked and corrected in line 333.
- Determination procedures of the activation energy and diffusion consistency coefficient have been mentioned in the section of ‘Material and methods’ and the outcomes and resulted models have been presented in lines 421-438.
- The sentence has been written as shown in lines 446-448.

Reviewer 2 Report
In this study, the impact of ultrasound pretreatment on the drying process of Asian Seabass Fish Skin and its influence on energy consumption during the drying process were investigated. In my opinion, the topic covered is interesting, but the manuscript needs a significant improvement.
My general objection is the following:
In Section 2.7, it is mentioned that ANOVA was applied; however, it is not evident anywhere in the manuscript, even though it should have been applied in several instances. For instance, in the case of drying time, the influence of two factors - temperature and ultrasound pretreatment - needs to be examined using two-way ANOVA, and the results should be presented as Mean ± Standard Deviation. Additionally, it is essential to determine whether the difference in drying times is statistically significant. To achieve this, it was necessary to prepare all drying samples in triplicate, and the drying process was to be repeated three times. The same applies to the data presented in Figure 9.
Specific remarks:
Line 31-32: Define abbreviations; the abstract should be self-explanatory without reading the rest of the manuscript..
Line 135: Exact intervals in which the sample masses were measured should be provided.
Line 147: Use the usual parentheses for equations to distinguish them from references.
Line 219: Specific heat capacity of what?
Figures 2 and 3 depict identical data; only one shows absolute moisture content, while the other shows moisture content in percentage.
The data presented in Figures 7 and 9 should be displayed with error bars.
Line 360: The activation energy of which process should be specified.
Line 374: Check the sentence, specifically which samples are being referred to?
Line 428-430: Revise the sentence. All samples were dried to the same moisture content, so you cannot conclude that the US pretreatment contributed to lower moisture content or a higher percentage of moisture removal.
Author Response
Dear respected reviewer,
Your effort in reviewing this manuscript is highly appreciated and many thanks for your encouragement. The modifications have been done accordingly and for easy track, they are highlighted with turquoise color in the manuscript.
Comment#1
“In Section 2.7, it is mentioned that ANOVA was applied; however, it is not evident anywhere in the manuscript, even though it should have been applied in several instances. For instance, in the case of drying time, the influence of two factors - temperature and ultrasound pretreatment - needs to be examined using two-way ANOVA, and the results should be presented as Mean ± Standard Deviation. Additionally, it is essential to determine whether the difference in drying times is statistically significant. To achieve this, it was necessary to prepare all drying samples in triplicate, and the drying process was to be repeated three times. The same applies to the data presented in Figure 9.”
Response#1
Many thanks for your valuable comment. The comparison was conducted using Two-way ANOVA as appeared in line 338 and Table 3.
Comment#2
“Line 31-32: Define abbreviations; the abstract should be self-explanatory without reading the rest of the manuscript.”
Response#2
Abbreviations have been clarified and highlighted with yellow color in line. 26.
Comment#3
“Line 135: Exact intervals in which the sample masses were measured should be provided.”
Response#3
Thank you so much. Weights of samples were recorded in triplicate at regular intervals of time (15, 30 and 60 min) until the final moisture content of 0.07 (d.b). It was inserted in line (164).
Comment#4
“Line 147: Use the usual parentheses for equations to distinguish them from references.”
Response#4
The parentheses have been modified for the better understanding.
Comment#5
“Line 219: Specific heat capacity of what?”
Response#5
Sorry. The equation used for determining the energy consumption has been replaced by Equation (13) in section 2.7. Specific heat capacity of air has been deleted.
Comment#6
“Figures 2 and 3 depict identical data; only one shows absolute moisture content, while the other shows moisture content in percentage.”
Response#6
Figure 2 represented the relationship between moisture content (d.b) of which unit is (g/g dry matter) vs time. However, Figure 3 displayed the moisture ratio (%).
Comment#7
“The data presented in Figures 7 and 9 should be displayed with error bars.”
Response#7
Error bars have been added in Figures 7 and 9.
Comment#8
“Line 360: The activation energy of which process should be specified.”
Response#8
The activation energy was for ‘drying process’. It has been specified in line 432.
Comment#9
“Line 374: Check the sentence, specifically which samples are being referred to?”
Response#9
The samples have been specifically clarified in lines 346-348 .
Comment#10
“Line 428-430: Revise the sentence. All samples were dried to the same moisture content, so you cannot conclude that the US pretreatment contributed to lower moisture content or a higher percentage of moisture removal.”
Response#10
Authors totally agree with the reviewer. It was a systematic error. The error has been rectified in conclusion section.

Reviewer 3 Report
A major revision of this paper is needed:
l. 31: explain appreviations, when used first
ll.50-67: very general claims are supported by only very specific references [5-13]
l. 87: is [25] really the right reference?
l. 110: is freezing of the skin necessary? what are the changes by freezing and thawing and the what the influences on later drying?
ll. 109-123: Sonication: What is the energy input (not us output) of the treatment? How much of the US output power is absorbed by the samples?
ll.134-138: Here the moisture content is specified as wet basis, lateron (l.138 drying rate) and (chapter 3) as dry basis! What is right ?
l.137: what is the water activity of the dried product? Is it by this shelf stable?
ll. 150-155: The equilibrium moisture content is dependent on the temperature and the air humidity. What values have been used here, and what is the actual value of the equilibium moisture content? Can it be really neglected, here?
table 1:
-in order to compare k-values (of different n) in the Page model use modified Page: MR=exp(-(k*t)^n)
-Vega-Lemus is not converging for large times
(sometimes the authors use * and sometimes . as multiplication sign)
chapter 2.4.1.
This is another drying model and should be listed in table 1.
equation 3: in the first denominator an "r" is missing.
equation 6: model is the same as Henderson and Pabis with a=8/pi²
l. 204: T must be the absolute temperature in K
equation 11 and 12: when T is the absolute temperature then (+273.15) can be omitted
l. 209: the physical unit of R is J/mol/K
eq. 11 to 12: both sides have to be divided by (m²/s) before logarithmizing the equation
l.213/eq. 13: This is not the energy consumption of the dryer , it is only the energy needed for air heating !
ll. 220-221: equation 14 gives only the energy output (not input or consumption) of the ultrasonic device. The convection oven is not shown here.
ll.230-231: it is more interesting how much energy is needed for the evaporation of one kg water, in order to compare it with other publications.
ll. 288-293: US-SS did not achieved "consistently lower moisture contents (for 55 °C this is only the case for large times)!
ll. 310-321: only be R² the indication for Page to be the best model is very weak. Furthermore the models have different numbers of parameters (1-4)!!!
table 2: the physical units of the parameters are missing!
ll. 326-340: how good is the fit with the simplified diffusion model?
eq. 19 an 20: physical units are missing, what is Rg (in difference to eq. 11)!
chapter 3.4.: the specific (complete input) energy (for removing 1 kg of water) should be given at least additionally
figs. 10 and 11: Mentioning the magnification is ok, but the scaling should be clearly visible in the figures!
A major revision of this paper is needed prior to publication:
l. 31: explain appreviations, when used first
ll.50-67: very general claims are supported by only very specific references [5-13]
l. 87: is [25] really the right reference?
l. 110: is freezing of the skin necessary? what are the changes by freezing and thawing and the what the influences on later drying?
ll. 109-123: Sonication: What is the energy input (not us output) of the treatment? How much of the US output power is absorbed by the samples?
ll.134-138: Here the moisture content is specified as wet basis, lateron (l.138 drying rate) and (chapter 3) as dry basis! What is right ?
l.137: what is the water activity of the dried product? Is it by this shelf stable?
ll. 150-155: The equilibrium moisture content is dependent on the temperature and the air humidity. What values have been used here, and what is the actual value of the equilibium moisture content? Can it be really neglected, here?
table 1:
-in order to compare k-values (of different n) in the Page model use modified Page: MR=exp(-(k*t)^n)
-Vega-Lemus is not converging for large times
(sometimes the authors use * and sometimes . as multiplication sign)
chapter 2.4.1.
This is another drying model and should be listed in table 1.
equation 3: in the first denominator an "r" is missing.
equation 6: model is the same as Henderson and Pabis with a=8/pi²
l. 204: T must be the absolute temperature in K
equation 11 and 12: when T is the absolute temperature then (+273.15) can be omitted
l. 209: the physical unit of R is J/mol/K
eq. 11 to 12: both sides have to be divided by (m²/s) before logarithmizing the equation
l.213/eq. 13: This is not the energy consumption of the dryer , it is only the energy needed for air heating !
ll. 220-221: equation 14 gives only the energy output (not input or consumption) of the ultrasonic device. The convection oven is not shown here.
ll.230-231: it is more interesting how much energy is needed for the evaporation of one kg water, in order to compare it with other publications.
ll. 288-293: US-SS did not achieved "consistently lower moisture contents (for 55 °C this is only the case for large times)!
ll. 310-321: only be R² the indication for Page to be the best model is very weak. Furthermore the models have different numbers of parameters (1-4)!!!
table 2: the physical units of the parameters are missing!
ll. 326-340: how good is the fit with the simplified diffusion model?
eq. 19 an 20: physical units are missing, what is Rg (in difference to eq. 11)!
chapter 3.4.: the specific (complete input) energy (for removing 1 kg of water) should be given at least additionally
figs. 10 and 11: Mentioning the magnification is ok, but the scaling should be clearly visible in the figures!
Author Response
Dear respected reviewer,
Your valuable efforts in reviewing this manuscript are greatly appreciated. All corrections have been made accordingly and for easy track, they are highlighted with pink color in the manuscript.
Comment#1
“l. 31: explain appreviations, when used first”
Response#1
Abbreviations have been defined and highlighted in line 26.
Comment#2
ll.50-67: very general claims are supported by only very specific references [5-13]
Response#2
It is a background for drying process. Different references have been cited to provide the strong background.
Comment#3
- 87: is [25] really the right reference?
Response#3
Sorry. It was a systematic error. The error occurred during ordering the reference list. However, the reference has been corrected. Please see line 92.
Comment#3
“l. 110: is freezing of the skin necessary? what are the changes by freezing and thawing and the what the influences on later drying?”
Response#3
Fresh fish skins were imported from a local factory in frozen state and kept frozen in the lab freezer to prevent any deterioration until conducting the experiment.
The effect of freezing and thawing on the drying behavior was not studied as all samples were frozen and stored for the same time before use. However, our plan in the upcoming experiment is to study these important effects. Thanks for your valuable notification.
Comment#4
- 109-123: Sonication: What is the energy input (not us output) of the treatment? How much of the US output power is absorbed by the samples?
Response#4
The energy input, output and the effective power of the ultrasound device used for this experiment are 750, 600 and 150 W, respectively. It has been included in the text Please see lines 143-144.
Comment#5
ll.134-138: Here the moisture content is specified as wet basis, lateron (l.138 drying rate) and (chapter 3) as dry basis! What is right ?
Response#5
Actually the measured moisture content was in wet basis and then converted to dry basis which is used to calculate the drying rate and used for drawing the graphs. However, the procedure has been clarified and the units were united. It can be found in the lines 156-172.
Comment#6
l.137: what is the water activity of the dried product? Is it by this shelf stable?
Response#6
Water activities were determined according to the procedure presented in lines 177-179 and the results were listed in Table 3. Also, the discussion has been made in lines 346-359.
Comment#7
“ll. 150-155: The equilibrium moisture content is dependent on the temperature and the air humidity. What values have been used here, and what is the actual value of the equilibium moisture content? Can it be really neglected, here?”
Response#7
The moisture content at equilibrium (Me) can be disregarded for extended drying durations, as its value (0.02 g/g d.b) is comparatively small when compared to both the initial moisture content (1.6±0.12 g/g d.b) and the moisture content at a specific time according to what previously mentioned by Fang, et al. [1]. Coincidentally, we are working on the sorption isotherms of fish skin to investigate the effect of temperature and relative humidity on the equilibrium moisture content. Hopefully, these valuable results would be published in another report.
Comment#8
table 1:
-in order to compare k-values (of different n) in the Page model use modified Page: MR=exp(-(k*t)^n)
-Vega-Lemus is not converging for large times
(sometimes the authors use * and sometimes . as multiplication sign)
Response#7
- Modified Page was tried but it did not properly work. However, Page model adequately fit the data.
- Regarding Vega-Lemus model, the reference of Cruz, et al. [2] was utilized as the product was dried for more than 35 h and the drying behavior was investigated using this model.
- Sorry, * was typing error. It was modified to dot in all equations in Table 1.
Comment#9
chapter 2.4.1.
This is another drying model and should be listed in table 1.
Response#9
Authors totally agree with the reviewer. To the best of our knowledge, Fick's second law of diffusion is a theoretical and a general equation from which the semi-theoretical models are simplified. Table 1 contains only the semi-theoretical and empirical models on which the manuscript is built.
Comment#10
equation 3: in the first denominator an "r" is missing.
Response#10
Sorry. It was a typing error. The error has been corrected in Equation (3).
Comment#11
equation 6: model is the same as Henderson and Pabis with a=8/pi²
Response#11
They are the same models.
Comment#12
- 204: T must be the absolute temperature in K
Response#12
The equations (11 and 12) have been modified following the reviewer’s suggestion.
Comment#13
equation 11 and 12: when T is the absolute temperature then (+273.15) can be omitted
Response#13
Thanks for your observation. +273.15 has been omitted from the previous Equations.
Comment#14
- 209: the physical unit of R is J/mol/K
Response#14
The unit has been modified to be J/(mol.K). It can be found in line 249.
Comment#15
- 11 to 12: both sides have to be divided by (m²/s) before logarithmizing the equation
Response#15
Noted with thanks. The change has been made.
Comment#16
l.213/eq. 13: This is not the energy consumption of the dryer , it is only the energy needed for air heating !
Response#16
Thanks for your comment. The equations have been modified as shown in section 2.7, equation (13) and then the total consumption has been recalculated.
Comment#17
- 220-221: equation 14 gives only the energy output (not input or consumption) of the ultrasonic device. The convection oven is not shown here.
Response#17
The equations have been modified as it can be seen in section 2.7, Equation (14) and then the total consumption has been recalculated.
Comment#18
ll.230-231: it is more interesting how much energy is needed for the evaporation of one kg water, in order to compare it with other publications.
Response#18
Energy is needed for the evaporation of one kg water was calculated using Equation 16 and the results were presented in Table 3 and then compared with another work.
Comment#19
- 288-293: US-SS did not achieved "consistently lower moisture contents (for 55 °C this is only the case for large times)!
Response#19
Sorry. This paragraph deals with the possible explanation for the effect of the US on moisture content level. Nonetheless, it was observed that the dried fish skins at different treatments had roughly the same moisture contents at different drying periods. Thus, this paragraph has been omitted.
Comment#20
- 310-321: only be R² the indication for Page to be the best model is very weak. Furthermore the models have different numbers of parameters (1-4)!!!
Response#20
Actually, the selection of the optimal mathematical models for the drying data was based on several criteria, including the highest coefficient of determination (R2), the lowest Chi-square ?2 and the lowest root mean square error (RMSE). The amendment can be found in line 361-368.
Comment#21
table 2: the physical units of the parameters are missing!
Response#21
Thanks for your observation. The units have been included. It can be found in lines 395-396.
Comment#22
- 326-340: how good is the fit with the simplified diffusion model?
Response#22
The equations inserted on Figure 8 displayed the goodness of the fit with the simplified diffusion model (R2 > 0.92).
Comment#23
- 19 an 20: physical units are missing, what is Rg (in difference to eq. 11)!
Response#23
The physical units have been included in the paragraph which the activation energies and moisture diffusivities have been interpreted.
Sorry. It was typing error. The equations (19 -20) have been modified.
Comment#24
chapter 3.4.: the specific (complete input) energy (for removing 1 kg of water) should be given at least additionally
Response#24
Thanks for your comment. The energy required for removing 1 kg of water has been determined and inserted in Table 3.
Comment#25
figs. 10 and 11: Mentioning the magnification is ok, but the scaling should be clearly visible in the figures!
Response#25
Actually, the scale can be clearly visible in the figures in the first version uploaded to the journal. However, during the conversion to PDF, the resolution was probably reduced. Authors believe that the final production would provide the clear scaling. The scale in the image of the cross-section figures is 500 µm, and for the surface it is 10 µm. However, authors will try to upload the figures with enhanced resolution. Thank you for your understanding.
Round 2
Reviewer 2 Report
In my opinion, the manuscript has now been significantly improved and can be accepted for publication after minor revisions.
Specific remarks:
Line 192-193: It is still unclear within which time interval the mass was measured, as the data presented in Figures 2, 3, and 5 are inconsistent with the intervals provided here.
Table 3 should be presented before Table 2, preferably at the beginning of Section 3 where the discussion of the results shown in Table 3 commences.
In Figure 3, the "%" units next to "Moisture ratio" should be removed. It would be appropriate for "%" to be displayed if the Moisture ratio is shown within the range of 0-100%.
Author Response
General Comment:
“In my opinion, the manuscript has now been significantly improved and can be accepted for publication after minor revisions.”
Response:
Your effort in reviewing this manuscript is highly appreciated and many thanks are given for your encouragement. The modifications have been done accordingly and for easy track, they are highlighted with turquoise color in the manuscript.
Comment#1
“Line 192-193: It is still unclear within which time interval the mass was measured, as the data presented in Figures 2, 3, and 5 are inconsistent with the intervals provided here.”
Response#1
Many thanks for your valuable comment. The drying time intervals appearing in the figures were varied. However, the details have been given for clarification. Please see line 165-166.
Comment#2
“Table 3 should be presented before Table 2, preferably at the beginning of Section 3 where the discussion of the results shown in Table 3 commences.”
Response#2
As per your request, the tables have been moved at the right place. Thank you.
Comment#3
“In Figure 3, the "%" units next to "Moisture ratio" should be removed. It would be appropriate for "%" to be displayed if the Moisture ratio is shown within the range of 0-100%.”
Response#3
Thank you so much. The percentage has been removed.

Reviewer 3 Report
Please find enclosed the review with the still necessary changes.

Minor editing of English language required.
Author Response
Responses to the comments of Reviewer#3
General Comment:
“Please find enclosed the review with the still necessary changes.”
Response:
Your valuable efforts in reviewing this manuscript are greatly appreciated. All corrections have been made accordingly and for easy track, they are highlighted with yellow color in the manuscript.
Comment#2
Round 1: ll.50-67: very general claims are supported by only very specific references [5-13]
Round 2: “More general Refernces would be helpful”
Response#2
Round 1: It is a background for drying process. Different references have been cited to provide the strong background.
Round 2: Thank you for your valuable suggestion. More general references have been included to this section Please see lines (63, 70 and 75).
Comment#3
R1: “l. 110: is freezing of the skin necessary? what are the changes by freezing and thawing and the what the influences on later drying?”
R2: “Please mention the possible influences of the freezing and thawing in the paper”
Response#3
R1: Fresh fish skins were imported from a local factory in frozen state and kept frozen in the lab freezer to prevent any deterioration caused by microorganisms until conducting the experiment. Authors accepted that some changes could take place during freezing and thawing, such as tissue damages, protein denaturation, lipid oxidation, etc. However, the effect of freezing and thawing on such changes of skin and drying behavior was not studied. All samples were frozen and stored for the same time before use. However, our plan in the upcoming experiment is to study these important effects. Thanks for your valuable notification.
R2: Freezing and thawing can determine the drying behavior and drying efficacy in both negative and positive ways. The aggregation of proteins and lipid oxidation might bring about the dense structure of skin during freezing/thawing, thus lowering the water evaporation. On the other hand, tissue damage caused by ice crystals during freezing might favor the looser structure, thereby facilitating the evaporation of water from the skin.
The discussion has been included in the manuscript. Please see line 519-524.
Comment#4
R1: ll. 109-123: Sonication: What is the energy input (not us output) of the treatment? How much of the US output power is absorbed by the samples?
R2: What is the meaning of effective power, 150 W is not mentioned in the paper”
Response#4
R1: The energy input and output of the ultrasound device used for this experiment were 750 and 600 W, respectively. It has been included in the text. Please see lines. 145-146.
R2: The effective power was brought from the device`s specification guideline. In our work, the output power was used for calculation of energy consumption. The effective power of 150 W has been reported. Please see line 144-145.
Comment#7
R1: “ll. 150-155: The equilibrium moisture content is dependent on the temperature and the air humidity. What values have been used here, and what is the actual value of the equilibium moisture content? Can it be really neglected, here?”
R2:”0.02 is more than ¼ of the final moisture content, can be really disregarded.”
Response#7
R1: The moisture content at equilibrium (Me) can be disregarded for extended drying durations, as its value (0.02 g/g d.b) is comparatively small when compared to both the initial moisture content (1.6±0.12 g/g d.b) and the moisture content at a specific time according to the previous results mentioned by Fang, et al. [1].
Coincidentally, we are working on the sorption isotherms of fish skin to investigate the effect of temperature and relative humidity on the equilibrium moisture content. Hopefully, these valuable results would be published in another report.
R2: Authors do agree with the reviewer. If it is proportional to the initial moisture content, it will represent around 1.25 % of the initial moisture content. Therefore, it can be disregarded for this reason.
Comment#8
R1: table 1:
-in order to compare k-values (of different n) in the Page model use modified Page: MR=exp(-(k*t)^n)
-Vega-Lemus is not converging for large times
(sometimes the authors use * and sometimes . as multiplication sign)
R2:
- “This cannot be true. Page and modified page can be transferred in each other”
- Nevertheless, Vega-Lemus is not converging for large times. It will grow or fall for infinity”
Response#7
R1:
- Modified Page was tried but it did not properly work. However, Page model adequately fit the data.
- Regarding Vega-Lemus model, the reference of Cruz, et al. [2] was utilized when the product was dried for more than 35 h and the drying behavior was investigated using this model.
- * was the typing error. It has been modified to ‘dot’ in all equations appearing in Table 1.
R2: Thanks. Modified page model was used. The outcomes can be found in Table 3. However, authors still want to keep Vega-Lemus model for comparison purpose.
Comment#10
R1: equation 3: in the first denominator an "r" is missing.
R2: “Still wrong: d2M/dr2”
Response#10
R1: Sorry. It was a typing error. The error has been corrected in Equation (3).
R2: Sorry, it has been corrected in line 208.
Comment#11
R1: equation 6: model is the same as Henderson and Pabis with a=8/pi²
R2:” Why not mentioned in paper”
Response#11
R1: They are the same models.
R2: It has been mentioned in equations (5, 6 and 7).
Comment#15
R1: eq. 11 to 12: both sides have to be divided by (m²/s) before logarithmizing the equation
R2: “no, cannot be seen in eq. 12.”
Response#15
R1: Noted with thanks. The change has been made.
R2: m2/s has been previously used in Eq. 9, line 234.
Comment#16
R1: l.213/eq. 13: This is not the energy consumption of the dryer, it is only the energy needed for air heating !
R2:” It stays absolutely unclear”, What power have been used for calculation”
Response#16
R1: Thanks for your comment. The equations have been modified as shown in section 2.7, equation (13) and then the total consumption has been recalculated.
R2: 600 W has been used for calculating the energy consumption of the sonication device.
Comment#17
R1: ll. 220-221: equation 14 gives only the energy output (not input or consumption) of the ultrasonic device. The convection oven is not shown here.
R2:” It stays absolutely unclear”, What power have been used for calculation”
Response#17
The equations have been modified as it can be seen in section 2.7, Equation (14) and then the total consumption has been recalculated.
R2: 1.6 kW has been used for calculating the energy consumption of the oven dryer. It is mentioned in line 258.
Comment#18
R1: ll.230-231: it is more interesting how much energy is needed for the evaporation of one kg water, in order to compare it with other publications.
R2: Omit numbers in Equations like 3600 to converge h to s>
Response#18
R1: Energy required for the evaporation of one kg water from the fish skin was calculated using Equation 16 and the results were presented in Table 2 and then compared with another works in the discussion part. Please see line 445-456.
R2: It has been omitted with thanks. It can be found in Eq. 15.
Comment#21
R1: table 2: the physical units of the parameters are missing!
R2: Units are included, but are wrong for page: if n >1 then k has not the unit of 1/h but 1/h^n , that`s why modified page should be used.
Response#21
R1: Thanks for your observation. The units have been included. It can be found in lines 415-416.
R2: Thanks. Modified page model has been used following the reviewer’s suggestion. The outcomes can be found in Table 3.
Comment#22
R1: ll. 326-340: how good is the fit with the simplified diffusion model?
R2: no, only the goodness of fit of the Arrhenius equation can be seen, but not of drying of diffusion model.
Response#22
R1: The equations inserted on Figure 8 displayed the goodness of the fit with the simplified diffusion model (R2 > 0.92).
R2: The simplified diffusion model is highly similar to the Henderson and Pabis model, as we previously discussed in our initial exchange. Consequently, the extent to which it accurately represents the experimental data closely mirrors the accuracy of the Henderson and Pabis model. This is evident from the data presented in Table 2. The R2 values for the Henderson and Pabis model were found to vary within the range of 0.981 to 0.998. The X2 values exhibited a range of 0.000 to 0.003, while the RMSE values displayed variations from 0.012 to 0.022.
Comment#23
R1: eq. 19 and 20: physical units are missing, what is Rg (in difference to eq. 11)!
R2: In the equations the units are still missing
Response#23
R1: The physical units have been included in the paragraph, in which the activation energies and moisture diffusivities have been interpreted. Sorry. It was typing error. The equations (19 -20) have been modified.
R2: Equations 21-22 are the outcomes of equation 11. So, activation energy value in J/mol converted to kJ/mol in the results. However, these values have been modified in the equations to be inconstant with the equation 11.
Comment#24
R1: chapter 3.4.: the specific (complete input) energy (for removing 1 kg of water) should be given at least additionally
R2: ok, but please compare it also to the heat of vaporization of water (app 2.2 MJ/kg, there is a large gap for energy savings.
Response#24
R1: Thanks for your comment. The energy required for removing 1 kg of water has been determined and inserted in Table 3.
R2: This large gap between the values of SEC and latent heat of vaporization is due to the fact that energy needed to evaporate water from a food system is more significant than evaporating water from the surface [3]. However, energy efficiency has been calculated using Eq. 16 and the results were presented in Figure 9b.
Comment#25
R: 1figs. 10 and 11: Mentioning the magnification is ok, but the scaling should be clearly visible in the figures!
R2: scales are still hard to recognize
Response#25
R1: Actually, the scale can be clearly visible in the figures in the first version uploaded to the journal. However, during the conversion to PDF, the resolution was probably reduced. Authors believe that the final production would provide the clear scaling. The scale in the image of the cross-section figures is 500 µm, and for the surface it is 10 µm. However, authors will try to upload the figures with enhanced resolution. Thank you for your understanding.
R2: The authors did our best to clarify the resolution of the images. The images now are well visualized in MS Word Version.
- Fang, M.; Huang, G.-J.; Sung, W.-C. Mass transfer and texture characteristics of fish skin during deep-fat frying, electrostatic frying, air frying and vacuum frying. LWT 2021, 137, 110494.
- Cruz, A.C.; Guiné, R.P.; Gonçalves, J.C. Drying kinetics and product quality for convective drying of apples (cvs. Golden Delicious and Granny Smith). International Journal of Fruit Science 2015, 15, 54-78.
- EL-Mesery, H.S.; El-Khawaga, S. Drying process on biomass: Evaluation of the drying performance and energy analysis of different dryers. Case Studies in Thermal Engineering 2022, 33, 101953.
